# Accelerator for Agglomeration in Sequencing Economics: "Leased" Industrial Zones

## Akifumi Kuchiki

Institute for International Trade and Investment, Tokyo 104-0045, Japan; akichan8107@ab.cyberhome.ne.jp

**Abstract:** This paper identifies the importance of reducing fixed costs for establishing industrial zones as part of an agglomeration policy. China's economic growth has been driven by the agglomeration of manufacturing firms via industrial zones that attract foreign direct investment. This investment enables the export of products by importing intermediate capital goods. According to the new trade theory of spatial economics, the number of firms in an agglomeration is inversely proportional to the fixed costs. The main accelerator of agglomeration after the master switch is the formation of segments that reduce firms' fixed costs. Via a factor analysis of manufacturing agglomeration segments in sequencing economics, this paper finds that "leased" industrial zones are accelerator segments in the formation process of manufacturing agglomerations.

**Keywords:** accelerator segment; leased industrial zones; sequencing economics; manufacturing agglomeration; fixed costs; spatial economics

**JEL Classification:** L22; O21; R11

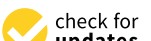



## 1. Introduction

Industrial agglomeration policy refers to the creation of agglomerations via policy. According to Fujita et al. (1999), agglomeration means the clustering of economic activities that are created and maintained via some form of circular logic.

Industrial agglomeration policies have been adopted in East Asia since the 1980s. The development of industrial zones brings together economic agglomeration and industrial clusters of economic activity. The prototype of industrial zones in Asia is the export processing zone concept in Kaohsiung, Taiwan, established in 1965. This model was then introduced in Penang, Malaysia, in the 1970s and in Tan Thuan, Ho Chi Minh City, in the 1990s.

Fujita et al. (1999) established spatial economics—or the study of where economic activity takes place and why. According to Oqubay and Lin (2020), the number of industrial zones in Asia has increased dramatically since the 1980s, with the Asia-Pacific region alone accounting for over 65% of global employment and exports. UNCTAD (2019) notes that in China, special economic zones (SEZs) have a strong positive effect on foreign direct investment (FDI), with SEZs accounting for more than 80% of cumulative FDI. Oqubay and Lin (2020) showed, via a dynamic approach, that understanding industrial hubs is important for the production-centred paradigm.

As of 2018, there are approximately 6000 existing and planned industrial parks worldwide, nearly 90% of which are in developing countries, with Asia accounting for nearly 70% and Africa for 5% (see UNCTAD (2019)). Oqubay (2020) noted that industrial parks in Ethiopia have played an insignificant role in the past but could play a larger role in the overall industrial development strategy in the future.

Pietrobelli (2020) analysed cluster development policies in Latin American countries as follows: cluster development policies in the Latin American subcontinent began to be implemented in the 2000s, and most of them were financed by international donor agencies,

including the Inter-American Development Bank; for example, cluster results for São Paulo and Minas Gerais, Brazil, show positive and significant effects on employment, export probability, and export levels.

Prior to the 1980s, Hirschman (1958) recommended fostering domestic industry by protecting domestic firms, but his strategy of unbalanced growth was introduced under the liberalisation of international trade and investment after the 1980s. In "The East Asian Miracle," the World Bank (1993) called export-led policies adopted in Asia via export processing zones the "export push strategy". Markusen (1996) classified industrial districts into five categories, including Marshall industrial districts and Italian-type industrial districts. Oqubay and Lin (2020), in the introduction to "*The Oxford Handbook of Industrial Hubs and Economic Development*," introduce the sequence of economics defined by Kuchiki, i.e., the flowchart approach to industrial agglomeration, and use it as the foundation for empirical and case study evidence obtained in Asia, Latin America, and Africa.[1] However, the flowchart approach lacked theoretical background and quantitative analysis.

Agglomeration segments are classified into four categories: physical infrastructure, institutions, human resources, and living environment, as shown in Table 1. Kanai and Ishida (2000) emphasised the importance of the cumulative process because, in spatial economics, the construction of any segment of agglomeration takes "time" in addition to space. Kuchiki and Sakai (2023) reflected on the analysis of the accumulation process as follows. First, Fujita and Kuchiki (2006) applied the flowchart method to the construction of the cumulative process, as shown in Figure 1. Second, Kuchiki (2020) proposed an architectural theory in the economy of sequence with respect to accumulation to find the optimal sequence for efficient segment construction. "Economies of sequence" in sequencing economics is defined as the ordering of any two segments in the set of segments that make up an agglomeration to efficiently construct that agglomeration. Third, Kuchiki and Sakai (2023) used the fact that spatial economics models derive segments that satisfy the symmetry-breaking condition to find that segments related to transport costs are the "master switch" for ordering segments of urban agglomerations. When a stable symmetric equilibrium is broken, the construction of the segments of an agglomeration equilibrium begins. However, no study has examined what the accelerator segment next to the master switch is when constructing the segments of a manufacturing agglomeration.

**Table 1.** China's (i) industrial agglomeration policy and (ii) industrial policy.

|  | Period 1: Start of (i) Industrial Agglomeration Policy (IAP) | Period 2: Start of (ii) Industrial Policy (IP) |
| --- | --- | --- |
| Period Classification | 1978–1984 Introduction of market economy | 1984–1992 Formation of market economy |
| Basic Idea | Elimination of supply shortage Industrial structure adjustment | Formation of unified markets (commodities, labour etc.) |
| Policy | **(i) "SEZ: Special Economic Zones 1979"** | **(i) ETDZ: State-level Economic and Technological Development Zones 1984** |
| Industry | Light industries by Township and Village Enterprises Reform | Basic industries |
|  | Textile Agriculture | Infrastructure Energy industry Steel and other materials industries |
| Regional Policy | **(i) "SEZ (Special Economic Zones): 4 locations" Shenzhen, Zhuhai, Xiantou, Xiamen (i)980** | (i) ETDZ: Dalian, Shanghai, Guangzhou, etc. 14 places (i) Economic Region in Southern China |
| Means | Direct control of quantity and price Allocation of capital and foreign currency System of distribution tickets of goods | **(i) ETDZ: FDI introduction policy** (ii) IP: Establishment of Industrial Policy Department 1988 (ii) IP: Announcement of the list of priority industries in 1989 |
| Special note | **(i) SEZ: Industrial agglomeration policy 1979 Establishment of Special Economic Zones** | (ii) IP: Merger and reorganization of enterprises |

Source: Prepared by Chen and Kuchiki (2000).

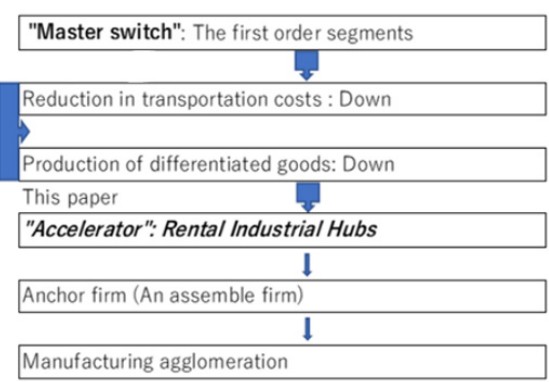

**Figure 1.** Manufacturing agglomeration policy.

The purpose of this paper is to find the accelerator segments that comprise an industrial agglomeration. An accelerator segment is defined as a segment that increases the number of firms in an industrial agglomeration.

Our research methodology is provided as follows. First, China is a country that has successfully pursued a policy of industrial agglomeration via the introduction of foreign capital. Therefore, we focus on the chronology of industrial agglomeration in China. The results of the Granger causality test for China reveal the process of industrial agglomeration formed by the introduction of foreign capital. We find that inward foreign direct investment (FDI) is associated with industrial agglomeration policy. Between 1987 and 2009 in China, the increase in the domestic investment to GDP ratio, according to Granger causality, resulted in rises in the rates of import, industrial, and GDP growth. Similarly, inward FDI, based on Granger causality, resulted in rises in the rates of import, industrial, and GDP growth. The inward FDI growth rate can explain the industrial growth rate and GDP growth rate, with the inward FDI growth rate having a positive regression relationship with both rates.

After the Plaza Accord of 1985, the exchange rates of countries such as Japan and South Korea were revalued. As domestic costs rose, these countries shifted their production bases overseas, and foreign direct investment in Asia surged. This paper focuses on the outward investment behaviour of Japanese firms. Using a study conducted by the Japan Bank for International Cooperation (JBIC), we use factor analysis to identify the factors that promote investment. The most promising factors are those that promote conditions for industrial agglomeration in each country. We find that factors related to incentives for foreign direct investment (FDI) include preferential tax policies for investment and stable policies to attract foreign investment.

Second, we apply a two-region model of the new economic geography in spatial economics to obtain a conclusion that the number of firms moving to a region is inversely proportional to the fixed costs of that region. This conclusion is the rationale for the fact that the accelerator segment is a fixed-cost factor of production.

Third, among the fixed costs, those related to the industrial agglomeration of labour-intensive manufacturing industries were identified via factor analysis using a survey conducted on Japanese companies by JETRO in 100 cities and regions in around 60 countries to determine investment-related costs when establishing operations in each city. We show that the rent of leased industrial zones was included in the same factor as the wages of workers in labour-intensive manufacturing industries. Therefore, we conclude that the establishment of leased industrial zones reduces fixed costs and increases the number of firms in industrial agglomerations.

As shown in Figure 1, the accelerator is the "leased" industrial zone, while the master switches of a manufacturing agglomeration policy are the reduction in transportation costs and the production of differentiated products.

The analysis in this paper suggests measures for successful industrial agglomeration policies. A consideration of the economies of sequence in sequencing economics is essential

to the implementation of such policies. The construction of accelerators will increase the efficiency of policy implementation. Further research in sequencing economics will be completed in the future.

The section structure is organised as follows. Section 2 explains the chronology of an industrial agglomeration policy. Section 3 describes the role of economic development zones as industrial zones from the perspective of spatial economics. Section 4 provides statistical analyses of industrial agglomeration policy. The factor analysis on investment-related costs in industrial zones is shown in Section 5, and the last section provides a summary and conclusions.

## 2. Chronology of the Chinese Industrial Agglomeration Policy

Here, the economic development zones as industrial zones include special economic zones (SEZs), economic and technological development zones (ETDZs), hi-tech industrial development zones (HIDZs), and pilot free-trade zones (PFTZs).[2]

The timeframe of an "industrial agglomeration policy" can be divided into seven main periods of the reform and opening-up policies, as shown in Tables 1–3.

**Table 2.** China's (i) industrial agglomeration policy and (ii) industrial policy.

|  | Period 3: Coexistence of (i) Industrial Agglomeration Policy and (ii) Industrial Policy | Period 4: Emphasis on (i) Industrial Agglomeration Policy |
|---|---|---|
| Period | 1992–1997 | 1997–2004 |
| Classification | **(i) Industrial agglomeration policy and (ii) Industrial policy** | (i) Industrial agglomeration policy |
| Basic Idea | Emphasis on international competition<br>Industrial structure rationalization | Globalization:<br>Competition with multinationals |
| Policy | **(i) HTDZ: State-level High-Tech Development Zones 1992: 52 locations**<br>**(ii) IP: Protection policy for infant industry** | **(i) "WTO Accession in 2001"** |
| Industry | **(ii) Industrial Policy Outline 1994: Four Major Pillar Industries**<br>**(ii) IP: Automobile Industry Policy 1994** | **(i) Emphasis on international competitiveness** |
|  | **(ii) IP: Four Major Pillar Industries: "Automobile, Machinery & Electronics, Petrochemicals, Construction"** | **(ii) Information Communication Industry**<br>New materials, Biotechnology |
| Regional Policy | **(i) "Yangtze River Economic Region"** | (i) Bohai Sea Rim Economic Region<br>(i) Western Great Development 2000 |
| Means | **(i) FDI introduction through development zones: ETDZ, HTDZ**<br>**(ii) Industrial policy** | **(i) Joint venture, technical cooperation**<br>**(ii) Merger and restructuring of enterprises** |
| Special note | **(i) IAP: Deng Xiaoping's "Southern Tour Speech 1992"**<br>**(i) IAP: Announcement of "Industrial Policy" Priority industries for foreign investment 1997**<br>**(ii) IP: 'Industrial Policy Outline' and 'Automobile Industry Policy' 1994** | **(i) Asian currency crisis 1997**<br>**(ii) Zhu Rongji:**<br>IP: Reform of state-owned enterprises 1998 |

Source: Prepared by Chen and Kuchiki (2000).

(1)   Establishment of "Special Economic Zones": the start of industrial agglomeration policy (1978–1984) (see Table 1).

The first period was dominated by a planned economy and the introduction of a market economy. A feature of this period was the establishment of "Special Economic Zones" in 1979 as a policy move toward the introduction of foreign direct investment (FDI) and industrial agglomeration. "Economic development zones," or (i) in Tables 1–3,

became centres of growth resulting from the introduction of foreign investment in industrial estates via preferential policies. There were three specific types (i): special economic zones (SEZs), state-level economic and technological development zones (ETDZs), and state-level high-tech industrial development zones (HIDZs).[3]

**Table 3.** China's (i) industrial agglomeration policy and (ii) industrial policy.

| | **Period 5: Emphasis on Harmonious Society** | **Period 6: New Start of (i) IAP and (ii) IP** | **Period 7: Integration of (i) IAP and (ii) IP** |
|---|---|---|---|
| Year | 2004–2010 | 2010–2013 Rebalancing | 2013–2022 |
| Classification | Harmonious Society | Change in pattern of economic development | Socialist Market Economy with Chinese Characteristics (i) (ii) |
| Basic<br>Ideology | Scientific View of Development<br>Passing the "turning point" | Escape from the "Trap of Middle-income Countries" | **(i) (ii) Domestic Circulation: Target 2035** |
| Policy | Reduction of disparities<br><br>Environmental conservation | **(i) PFTZ: Shanghai Pilot Free Trade Zone 2013**<br>**(ii) IP: Strategic Emerging Industries 2010** | **(i) (ii) SEI: Strategic Emerging Industries**<br>**(i) (ii) PFTZ: Beijing Pilot Free Trade Zone 2021**<br>One Belt, One Road Joint Construction 2013 |
| Industry | **Upgrading industrial structure:**<br><br>**(ii) High-tech industries**<br><br>**(ii) Biomedicine**<br><br>**(ii) High-tech informatization** | **(ii) IP: 7 major industries:**<br><br>**Environmental protection,**<br><br>**Information Communication, New energy, New energy automobiles, etc.** | **(i) IP: Modern Service Industries 2013**<br>**(ii) IP: Strategic Emerging Industries**<br>Post COVID-19<br>Green economy and Digital economy |
| Regional | (i) Western Development | (i) Northeast Regional Development Plan (2016–2020) | **(i) (ii) PFTZ: 21 Pilot Free Trade Zones 2021** |
| Means | **(ii) Creation of independent technology** | New rural construction | (i) (ii) New infrastructure construction 2020 |
| Special note | Increase in minimum wage<br><br>Three-farm problem<br><br>Lehman Shock 2008 | Reduction of corruption<br><br>Emphasis on green industry<br><br>China's GDP 2nd in the world 2010 | **(i) (ii) 1st in the world: Fortune's World 500 largest companies:**<br><br>124 companies 2020 |

Source: Prepared by Chen and Kuchiki (2000).

First, SEZs served as pilot zones for economic reforms and encouraged the introduction of foreign capital as part of the policy of opening up to the outside world via preferential systems. These preferential systems included preferential taxation, such as corporate tax exemptions and tariff exemptions for equipment imports, as well as preferential treatment in terms of management autonomy and foreign currency management for foreign companies. The cities designated in 1980 were Shenzhen, Zhuhai, and Shantou in Guangdong and Xiamen in Fujian. Guangdong was identified as a region-specific SEZ, while ETDZs and HIDZs were later adopted in other regions, as explained below.

Second, ETDZs aimed to further open China up to the outside world, following the lead of SEZs in 1984, by granting the same preferential policies as those of SEZs. The State Council designated 14 such zones in 12 coastal open cities.[4]

Third, in the second period, HIDZs were implemented in 1988 with the aim of developing emerging industries. The Beijing Hi-Tech Industrial Development Zone was approved by the State Council and is the predecessor of the "Zhongguancun Science and Technology Park". This led in part to the establishment of the "Beijing Pilot Free Trade Zone" in 2020 (see State Council of China (2020a)).

Hence, the special economic zone approach became the prototype for China's "industrial agglomeration policy".

(2)   The beginning of industrial policy (1984–1992) (see Table 1).

The second period, from 1984 to 1992, was a transitional period between the first period, characterised by a planned economy, and the third period, characterised by a market economy. During this period, China developed its market economy and formed a "unified market".

(3)   The coexistence of industrial agglomeration policy and industrial policy (1992–1997) (see Table 2).

The third period, from 1992 to 1997, saw the coexistence of two economic policies: (i) the policy of introducing foreign capital and industrial agglomeration and (ii) the policy of protecting domestic and infant industries.

In 1992, Deng Xiaoping delivered his "Southern Tour Speech," calling for reform and promoting an opening-up of the economy and policies of industrial agglomeration via introducing FDI. The policy was implemented in a number of coastal cities, starting with the Shanghai Pudong development.[5]

After Shanghai, the development mechanism based on the industrial agglomeration policy via the introduction of FDI saw subsequent success in Tianjin and Chongqing[6] High-tech enterprises are knowledge- and technology-intensive enterprises that utilise advanced technology. The Beijing HTDZ (a state-level high-tech industrial development zone) was approved in 1988 and was the precursor of the Zhongguancun Science and Technology Park. The number of HTDZs thereafter increased to 27 in 1990 and 52 in 1992.

(4)   Industrial agglomeration policy with an emphasis on international competitiveness of enterprises (1997–2004) (see Table 2).

In preparation for its "World Trade Organisation (WTO) Accession" in 2001, China modified its domestic legal system by enacting the Anti-Monopoly Law and amending the Foreign Trade Law and the Export Commodities Inspection Law, among others, to improve the investment environment. In addition to the successful South China Economic Region, which had previously been centred on Guangdong and Shenzhen, this led to the introduction of foreign capital into the Yangtze River Economic Region, including Shanghai, and later the Beijing–Tianjin–Hebei Economic Region, including the Beijing, Tianjin, and Hebei provinces. This policy resulted in China's industrial agglomeration.

(5)   Scientific View of Development after the turning point (2004–2010) (see Table 3).

China passed the turning point around 2004.[7] The country managed its economy properly, achieved high economic growth via the Five-Year Plan and the National People's Congress, and changed its development pattern by focusing on "Scientific Development".[8]

(6)   "Strategic emerging industries" as an industrial policy to escape from the middle-income-country trap (2010–2013) (see Table 3).

China sought to escape from the "middle-income country trap" during this period.
Regional industrial agglomeration policies were further expanded, and the emphasis shifted to the Great Western Development policy centred on the Chengdu–Chongqing Economic Zone and the promotion of the Northeast region.

(7)   Industrial agglomeration policy via "pilot free trade zones" (2013–2022) (see Table 3).

Pilot free-trade zones are the basis for the formation of industrial "agglomerations". The first pilot free-trade zone was established in Shanghai in 2013 to explore new growth in tertiary industries. By attracting foreign capital in consumer-related service industries, China aimed to develop (i) "modern service industries".[9] In 2017, the focus was expanded from "Modern Service Industries" to "advanced manufacturing industries". In 2020, pilot free-trade zones were established in three provinces—Beijing, Hunan, and Anhui—bringing the total to 21 (see State Council of China (2020b)).[10]

As described above, following the reform and opening-up policies, China has continued to implement industrial agglomeration policies since 1978. In the next section, we will show that the industrial agglomeration formed has led to China's economic growth. We apply factor analysis to identify segments that should be of priority in constructing labour-intensive manufacturing agglomerations.

### 3. The Role of Economic Development Zones as Industrial Zones

Table 4 shows the segments of the economic development zones. The segments related to transportation costs consist of infrastructure, such as transportation; institutions, such as one-stop services; and human resources, such as skilled labour. Figure 1 illustrates the role of economic development zones as master switches in introducing industrial agglomeration policies. The master switches in the agglomeration policy are reductions in transportation costs and initiating the production of differentiated goods with low values of elasticity of substitution between any two types of goods and services. This will be clarified below using a theoretical model of spatial economics.

**Table 4.** The segments of economic development zones.

| | **Segments** | |
|---|---|---|
| Capacity | 1. Infrastructure | (1) Water |
| | | (2) Electricity |
| | | (3) Communication |
| | | (4) Transport (Transport costs) |
| | 2. Institutions | (1) One-stop services (Transport costs) |
| | | (2) Deregulation |
| | | (3) Preferential treatments (tax incentives, etc.) (Transport costs) |
| | | (4) Laws and regulations (bankruptcy laws and intellectual property right) |
| | 3. Human resource | (1) Unskilled labor |
| | | (2) Skilled labor |
| | | (3) Professionals |
| | 4. Living conditions | (1) Housing |
| | | (2) International schools |
| | | (3) Hospitals |
| | | (4) Entertainment |

Source: Prepared by Kuchiki.

Helpman and Krugman (1985) provided a new trade theory in spatial economics, in which the equilibrium number of firms is derived based on a general equilibrium model (this model is described in "The model" in Appendix B). The economy consists of two countries, 1 and 2. In this model, the two sectors are the manufacturing sector and the agricultural sector, the population, $L_k$ of country k, is constant, and the Cobb–Douglas utility function is used. The model adopts the Dixit and Stiglitz (1977) monopolistic competition model framework and assumes that many firms produce a variety of differentiated goods in both country 1 and country 2.

The model assumes that there is free entry and exit of firms based on profits and losses. Thus, based on the zero-profit condition, the number of firms is

$$n_k = (\mu/\sigma)[y_k L_k/(F_k - \varphi F_s) + \varphi y_s L_s/(\varphi F_k - F_s)], \; k = 1, 2, \; s = 1, 2, \; s \neq k,$$

where $L_k$ is the population number, μ is the elasticity of differentiated goods in the Cobb–Douglas utility function, σ is the elasticity of substitution between any two of the varieties of goods, $\varphi \equiv \tau^{(1-\sigma)}$, τ is the "iceberg" form of transport costs, and $F_k$ is the fixed cost. We obtain the following equations:

$$\partial n_k / \partial F_k = -(\mu/\sigma) \, [y_k L_k / (F_k - \varphi F_s)^2 + \varphi^2 \, y_s L_s / (\varphi F_k - F_s)^2] < 0,$$

$$\partial n_k / \partial F_s = (\mu\varphi/\sigma) \, [y_k L_k / (F_k - \varphi F_s)^2 + y_s L_s / (\varphi F_k - F_s)^2] > 0.$$

Thus, the above equations reveal that the number of firm agglomerations is inversely proportional to the fixed cost $F_k$ and that reducing fixed costs, such as by establishing leased industrial bases, leads to an increase in the number of firm agglomerations.

## 4. Statistical Analyses of Industrial Agglomeration Policy

This section statistically examines the processes of (i) foreign direct investment (FDI) agglomeration, (ii) intermediate goods import, (iii) increases in industrial output, and (iv) growth in GDP. China's economic growth involved importing capital goods and raw materials, producing products, and exporting them. This can be seen in the statistics in Tables 5 and 6.

**Table 5.** Structure of export and import (Unit: million$,%).

|  | 1997 | | 1998 | | 1997 | | 1998 | |
|---|---|---|---|---|---|---|---|---|
|  | **Export** | **Share** | **Export** | **Share** | **Import** | **Share** | **Import** | **Share** |
| Machinery, Transport equipment Machinery: machinery, appliances, electrical equipment and parts, recorders and playback equipment, equipment for recording and reproducing video images and sound, and parts and accessories. | 488.16 | 26.7 | 564.21 | 30.7 | 578.66 | 40.6 | 620.89 | 44.3 |
|  | 382.7 | 20.9 | 436.29 | 23.7 | 467.58 | 32.8 | 509.09 | 36.3 |
| Vehicles: Aircraft, ships and related transport equipment | 52.73 | 2.9 | 63.96 | 3.5 | 55.54 | 3.9 | 55.9 | 4.0 |
| Railway and tramway locomotives, rolling stock and parts, railway and tramway fixtures and accessories, various mechanical (electrical), traffic signalling equipment | 11.95 | 0.7 | 18.21 | 1.0 | 1.19 | 0.1 | 2.24 | 0.2 |
| Vehicles and their parts and accessories, excluding railways and trams. | 21.57 | 1.2 | 22.72 | 1.2 | 18.96 | 1.3 | 20.03 | 1.4 |
| Aircraft, spacecraft and parts thereof | 2.91 | 0.2 | 4.4 | 0.2 | 32.35 | 2.3 | 31.75 | 2.3 |
| Ships and floating structures | 16.3 | 0.9 | 18.63 | 1.0 | 3.04 | 0.2 | 1.88 | 0.1 |
| Optical, photographic, cinematographic, measuring, testing, medical or surgical instruments, precision instruments and apparatus, clocks and watches, musical instruments, and parts and accessories thereof | 63.21 | 3.5 | 65.64 | 3.6 | 46.72 | 3.3 | 49.26 | 3.5 |
| Total | 1827.92 | 100.0 | 1837.57 | 100.0 | 1423.7 | 100.0 | 1401.66 | 100.0 |

Source: Prepared by Kuchiki based on (National Bureau of Statistics of China 2008, 2019).

**Table 6.** Structure of export and import (Unit: million $,%) (Export(FOB)).

| Classification | 2008 Value | Share | 2009 Value | Share | 2017 Value | Share | Value | 2018 Share |
|---|---|---|---|---|---|---|---|---|
| Industrial products | 1,352,736 | 94.5511 | 1,138,564 | 94.74903 | 2,145,813 | 94.79974 | 2,352,021 | 94.6 |
| Machinery, Transport equitment | 673,329 | 47.06314 | 590,427 | 49.13416 | 1,082,905 | 47.84159 | 1,208,055 | 48.6 |
| Textile, rubber and mineral products | 262,391 | 18.34013 | 184,775 | 15.37661 | 368,054 | 16.26024 | 404,753 | 16.3 |
| Chemicals and related products | 79,346 | 5.545984 | 62,048 | 5.163511 | 141,329 | 6.243765 | 167,525 | 6.7 |
| Miscellaneous products | 335,959 | 23.48226 | 299,670 | 24.93794 | 547,767 | 24.19976 | 565,814 | 22.7 |
| Unclassified other products | 1710 | 0.119522 | 1645 | 0.136894 | 5758 | 0.254382 | 5873 | 0.2 |
| Primary products | 77,957 | 5.448898 | 63,099 | 5.250973 | 117,709 | 5.200259 | 135,086 | 5.4 |
| Total | 1,430,693 | 100 | 1,201,663 | 100 | 2,263,522 | 100 | 2,487,401 | 100 |

|  |  |  |  |  |  |  | Import (CIF) | |
|---|---|---|---|---|---|---|---|---|
| Classification | 2008 Value | Share | 2009 Value | Share | 2017 Value | Share | 2018 Value | Share |
| Industrial products | 770,167 | 68.00219 | 716,353 | 71.2 | 1,263,918 | 68.65456 | 1,434,025 | 67.1 |
| Machinery, Transport equitment | 441,765 | 39.00581 | 407,999 | 40.6 | 734,846 | 39.91598 | 839,524 | 39.3 |
| Textile, rubber and mineral products | 107,165 | 9.462175 | 107,732 | 10.7 | 135,075 | 7.337117 | 151,452 | 7.1 |
| Chemicals and related products | 119,188 | 10.52375 | 112,124 | 11.2 | 193,744 | 10.52395 | 233,683 | 10.9 |
| Miscellaneous products | 97,641 | 8.62125 | 85,192 | 8.5 | 134,175 | 7.28823 | 143,759 | 6.7 |
| Unclassified other products | 4409 | 0.389294 | 3306 | 0.3 | 66,079 | 3.589334 | 75,607 | 3.5 |
| Primary products | 362,395 | 31.99781 | 289,202 | 28.8 | 577,064 | 31.34544 | 701,613 | 32.9 |
| Total | 1,132,562 | 100 | 1,005,555 | 100 | 1,840,982 | 100 | 2,135,637 | 100 |

Source: Prepared by Kuchiki based on (National Bureau of Statistics of China 2008, 2019).

Meanwhile, regarding exports, the contribution of machinery and transport equipment to total exports rose from 26.7% in 1997 to 47.1% in 2008. This stayed at the same level in 2017, at 47.8 %. On the other hand, regarding imports, the contribution of machinery and transport equipment to total imports was 40.6% in 1997, 39% in 2008, and the same in 2017, at 39.9%. Machinery is used here to refer to machinery, appliances, electrical equipment and parts, recorders and playback equipment, and equipment for recording. We hereafter examine whether these imports have led to industrial growth and subsequent GDP growth.

As shown in Tables 7 and 8, we tested Granger causality and correlations between the domestic investment rate as a percentage of GDP, inward FDI, and imports (See "Table A1. Data from Granger Causality Analysis" in Appendix A).

The results of the Granger causality tests in Table 7 show that, between 1987 and 2009, the increase in the domestic investment/GDP ratio caused an increase in the import growth rate with a 1-year lag (the data used are presented in Appendix A). The increase in import growth rate led to an increase in industry growth rate with a 5-year lag, and the increase in industry growth rate led to an increase in the GDP growth rate with a 3-year lag. Therefore, based on Granger causality, the increase in the domestic investment/GDP ratio caused a rise in the rates of import, industrial, and GDP growth, in that order, with domestic investment having a positive effect on GDP growth in the first period examined.

**Table 7.** Granger causality tests on domestic investment and foreign direct investment.

| Hypothesis | Lag Years | F-Test | *p*-Value | Period |
|---|---|---|---|---|
| (i) domestic investment—GDP ratio causes (x) export growth rate | 3 | 3.186 | 0.03905 * | 1987–2009 |
| (i) domestic investment—GDP ratio causes (g) industy growth rate | 2 | 5.7014 | 0.007324 *** | 1987–2009 |
| (i) -> (x), (i) -> (g) | | | | |
| (f) foreign direct investment causes (m) import growth rate | 2 | 3.5913 | 0.0346 * | 1987–2018 |
| **(f) -> (m) -> (g) -> (y) (1987–2009)** | | | | |

Source: Prepared by Kuchiki. * Significant at the 5 percent level. *** Significant at the 0 percent level.

**Table 8.** Linear regression on growth rate of foreign direct investment.

| | Coefficients | *p*-Value | Adjusted R-Squared | Period |
|---|---|---|---|---|
| (g) industry growth rate | | | | 1987–2018 |
| Intercept | 9.23654 | $4.61 \times 10^{-16}$ *** | | |
| (a) growth rate of foreign direct investment | 0.08552 | $2.24 \times 10^{-6}$ *** | | |
| | | | 0.5157 | |
| (y) GDP growth rate | | | | 1987–2018 |
| Intercept | 8.379637 | $2.750743 \times 10^{-20}$ *** | | |
| (a) growth rate of foreign direct investment | 0.040663 | 0.000148 *** | | |
| | | | 0.3655 | |
| **(Δf/f) = (a) <-> (g) <-> (y)** | | | | |

Source: Prepared by Kuchiki (See Appendix A Data Table). *** Significant at the 0 percent level. Note: a(n) = 100 × [f(n) − f(n − 1)]/f(n − 1).

Between 1987 and 2018, inward FDI caused import growth with a two-year lag. Thus, between 1987 and 2009, similarly to the case of the investment/GDP ratio, FDI caused a rise in the rates of import, industrial, and GDP economic growth. In the second period, FDI also had a positive effect on GDP growth.

The results of the regression analyses in Table 8 show that the FDI growth rate significantly regressed with the industry growth rate at a significance level of 0%, which also significantly regressed with the GDP growth rate at a significance level of 0%. Thus, it can be reconfirmed that FDI has had a positive effect on GDP economic growth.

The above findings show that from 1987 to 2009, both domestic investment and FDI were effective in promoting GDP growth until period 5.

## 5. Factor Analysis

Factor analysis is used to extract common factors latent behind observed variables. Here, the observed data are the explained variable $x_i$, the common factor is the explanatory variable f, and the part that cannot be explained by the common factor is the error term $u_i$, which is the unique factor. The coefficient $b_{ij}$ of the explanatory variable that represents the common factor is the factor loading ($x_i = b_{ij} f + u_i$); here, the factor loadings are obtained by multiplying the square root of the eigenvalues of the factor loading matrix by the eigenvector.

The number of common factors is determined, and each common factor is interpreted in terms of the factors it has in common. In this paper, manufacturing labour wages and industrial park rents are included in the same common factor. This implies that they are correlated in the formation of industrial agglomeration in the manufacturing sector.

(1)    FDI-led agglomeration as an incentive for foreign direct investment

JBIC (2022) conducted a study to examine in detail the main drivers of investment that countries experience, presented in the *Report on Survey of Overseas Business Expansion of Japanese Manufacturing Companies*. The top 10 most promising countries from 2007 to 2022 include the following six: India, Vietnam, Indonesia, Thailand, the U.S., and China. The factor analysis in this paper presents the main drivers of investment.

The first factor, ML1, of promising factors, includes those that promote industrial agglomeration. As shown in Figure 2, factors related to reductions in transportation costs in the broadest sense include (o) well-developed local logistics services and (n) the development of local physical infrastructure. Physical infrastructure includes transportation, electricity, and telecommunications. Their factor loadings are 1.1 and 1, respectively (factor loadings are shown in parentheses below). One factor related to the institutional aspect of soft infrastructure is (r) stable political and social conditions (1). Others are (l) the profitability of the local market and (m) product development zones (0.9 and 0.7, respectively).

The above points indicate that there is investment potential in areas where industrial agglomeration is possible.

## Factor Analysis

**Figure 2.** FDI-led agglomeration. ML1: Agglomeration: (o) Local logistics services (1.1). (n) Local physical infrastructure (1). (r) Stable political and social conditions (1). (l) Profitability of local market (0.9). (m) A base for product development (0.7). ML2: Human Resources: (b) Low-wage labor (0.7) (a) Excellent human resources (0.5) (f) Risk diversification receptacle for other countries (0.6). ML3: Export processing zone: (p) Preferential tax incentives for investment (0.9). (q) Stable policies to attract foreign investment (0.9). (h) An export base to Japan (0.8). (g) An export base to third countries (0.6). (d) A supple base to assemble makers (0.4). ML4: Raw material procurement: (i) Advantage in procurement of raw materials (0.6) (c) Cheap parts and raw materials (0.5). (): Factor loading. Red line means insignificant factor. Source: author.

The third factor, ML3, related to incentives for foreign direct investment includes (p) preferential tax incentives for investment (system) and (q) stable policies to attract foreign investment (both 0.9). Factors related to export-processing zones are (h) an export base to Japan and (g) an export base to third countries (0.8 and 0.6, respectively). In addition, the second factor, ML2, relates to human resources: (b) cheap labour and (a) high-quality human resources. The fourth factor, ML4, relates to materials, i.e., (i) advantages in terms of raw material procurement and (e) low-cost parts and raw materials.

Hence, the first factor, ML1, represents "industrial agglomeration" and the "stability of policies to attract foreign investment."

(2)    Accelerators of agglomeration policies

The theory of spatial economics supports the conclusion that reduced fixed costs promote industrial agglomeration. Land and buildings account for a high proportion of the fixed costs occupied by foreign investment in the manufacturing sector. Therefore, the establishment of industrial zones reduces firms' fixed costs. In particular, "leased" industrial zones significantly reduce the fixed costs of the occupying companies compared to purchased industrial zones.

The Japan External Trade Organization (JETRO) surveys Japanese companies operating in 100 cities and regions in about 60 countries around the world to determine the investment-related costs of setting up operations in each city. This is shown in JETRO (2010; 2022) for companies' overseas expansion (data used are shown in "Table A2. Data for Factor

Analysis" in Appendix A). In conclusion, the relationship between "manufacturing workers (general engineering workers: W1)" and "leased industrial zones: Z2" is important for the introduction of foreign investment due to the reduction in fixed costs, as shown in "Table A3. Loadings of Factor Analysis" in Appendix A. Factor analysis indicates that the reduction in fixed costs is an accelerator for expanding the number of firms attracted to foreign countries.

The survey cities used in this paper are listed in Table 9. Table 10 also shows the segments of investment-related costs corresponding to the segments in Table 4. The investment-related costs in Table 10 include wages for manufacturing workers (W1), wages for engineers (W2), purchased rents for industrial zones (Z1), leased prices of industrial zones (Z2), commercial electricity rates (P1), and container transportation to Japan (C1).

**Table 9.** List of survey cities for factor analysis.

| | Survey Cities for Factor Analysis |
|---|---|
| China | 1. Chengdu |
| area | 2. Dalian |
| Year | 3. Guangzhou |
| 2021 | 4. Qingdao |
| | 5. Shanghai |
| | 6. Shinzhen |
| | 7. Wuhan |
| | 8. Chongqing |
| | 9. Beijing |
| | 10. Hong Kong |
| | 11. Taiwan |
| India | 12. Ahmedabad |
| 2022 | 13. Bengaluru |
| | 14. Chennai |
| | 15. Munbai |
| | 16. New Delhi |
| Asia | 1. Beijing |
| Year | 2. Shanghai |
| 2010 | 3. Guangzhou |
| | 4. Dalian |
| | 5. Shenyang |
| | 6. Qingdao |
| | 7. Shinzhen |
| | 8. Bangkok |
| | 9. Jakarta |
| | 10. Manila |
| | 11. Sebu |
| | 12. Bengaluru |
| | 13. Colombo |

**Table 9.** *Cont.*

| Survey Cities for Factor Analysis | | |
|---|---|---|
| Mexico | 17. Irapuato | |
| Year 2022 | 18. Mexico City | |
| | 19. Monterrey | |
| | 20. Queretaro | |
| | 21. San Luis Potosi | |
| | 22. Tijuana | |
| | 23. Aguascalientes | |
| South | 24. Asuncion | Paraguay |
| America | 25. Buenos Aires | Argentina |
| 2022 | 26. Campinas | Brasil |
| | 27. Manaus | Brasil |
| | 28. Rio de Janeiro | Brasil |
| | 29. San Paulo | Brasil |
| | 30. Santiago | Chile |
| ASEAN | 31. Bangkok | Thailand |
| 2022 | 32. Jakarta | Indonesia |
| | 33. Kuala Lumpur | Malaysia |
| | 32. Jakarta | Vietnam |
| | 35. Danang | Vietnam |
| | 36. Hanoi | Vietnam |
| | 37. Hochiminh | Lao |
| | 38. Yangon | Myanmer |
| | 39. Bientian | Cambosia |
| | 40. Punon Phen | Cambosia |

Source: Prepared by Kuchiki.

(2)-1 Fixed-cost reduction factors for manufacturing industrial agglomeration in 2010 (regions covered: Beijing, Shanghai, Guangzhou, Dalian, Shenyang, Qingdao, Shenzhen, Bangkok, Jakarta, Manila, Cebu, Bangalore, and Colombo).

The number of factors depends on the number of eigenvalues greater than or equal to one. As shown in Figure 3, the fourth factor of investment-related costs in 2010 consisted of "human resources"-related factors (W1, W2, and W3) for workers (general engineers), engineers (mid-level engineers), and middle managers, and "industrial zone"-related factors (Z1, Z2, and Z3) for industrial zone (land) purchase prices, industrial zone rental rates, and office rent. The relationship between "W1 and Z1, W1 and Z2" is particularly important. Cities with high scores in the four factors are Beijing, Shanghai, Shenzhen, and Bangkok.

**Table 10.** Investment-related costs.

| | Survey items | City: Dalian, China |
|---|---|---|
| | | **Survey Period: November 2022~January 2023 Exchange Rate: 1US$ = (1 November 2022, Interbank) Including VAT** |
| | Survey items | US$ |
| W1 | worker (general laborer) (per month) (manufacturing) | 506 |
| W2 | engineer (intermediate technitian) (per month) (same as above) | 822 |
| W3 | middle management (section chief) (per month) (same) | 1268 |
| W4 | staff (general office work) (per month) (non-manufacturing) | 1037 |
| W5 | manager (section chief) (per month) (same as above) | 2185 |
| Z1 | industrial zone (land) (purchase price) (per square meter) | 96 |
| Z2 | industrial zone rent (per square meter, per month) | 2.97 |
| Z3 | office rent (per squre meter, per month | 22 |
| P1 | commercial electricity rates (pre 1 kWh) | 0.095 |
| P2 | commercial water rates (per cubic meter) | 0.68 |
| P3 | commercial gas rates (per 1 kg) | 0.46 |
| C1 | container transport to Japan (40 ft) | 804 |
| C2 | container transport to the third country (40 ft) | 13,915 |

Source: Japan External Trade Organization (JETRO 2022).

## Factor Analysis

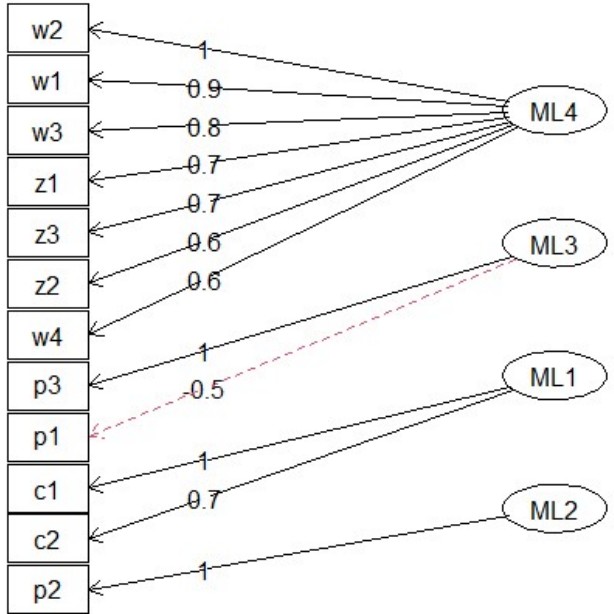

**Figure 3.** Investment cost comparison for 2010. Source: authors. Red line means insignificant factor.

The second factor is "gas prices," the third is "container transportation charges for exports to Japan and exports to third countries," and the fourth factor is "water prices".

(2)-2 Fixed-cost reduction factors for manufacturing industrial agglomerations in 2021–22.

The first factors of investment-related costs in 2021 are "human resources"-related factors (W1, W2, and W3) for workers (general engineering workers), engineers (mid-level technicians) and middle managers, and industrial zone rental rates (Z2). The relationship between W1 and Z2 is particularly important.[11] The cities with the highest scores for factor

1 are Shanghai, Beijing, Hong Kong, Taiwan, Buenos Aires, and Sao Paulo, as shown in "Table A4. High score cities of Factor Analysis" in Appendix A.

Manufacturing workers are related to both leased and purchased industrial zones, with the former being particularly effective in reducing fixed costs for firms occupying these zones.

Factor 2 is related to "human resources," namely the wages of office workers in the manufacturing industry and managers (section managers) in the non-manufacturing industry, as well as electricity and gas prices. Factor 3 is water rates, container transportation rates, and engineers' wages.

(2)-3 General engineering workers and leased industrial zones in the same factor.

The results of the factor analysis in Figure 4 showed that manufacturing worker wages and rental rates for leased industrial zones were included in the same factor, indicating that leased industrial zones contribute to increasing the agglomeration of manufacturing industries. The fact that the rental rates for leased industrial zones are included in the same factor suggests that these zones are important cost and economic factors for the manufacturing industry. The use of leased industrial zones allows manufacturing firms to engage in production activities without having to invest capital and provides flexibility and reduces risk.

## Factor Analysis

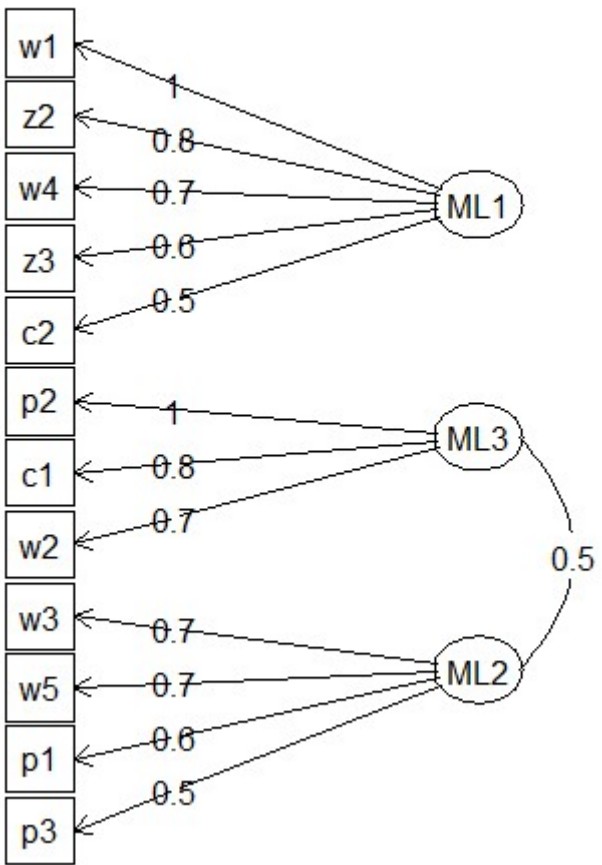

**Figure 4.** Investment cost comparison for 2021–22. Source: authors.

Therefore, it is effective to establish "leased" industrial zones (Z2) to utilise existing workers (W1) when building manufacturing agglomerations.

## 6. Summary and Conclusions

The segment that satisfies the breaking condition represents the master switch in the first sequence of the agglomeration policy. Following the master switch, the construction of the main driver segment is the next stage in the agglomeration sequence. The agglomeration of manufacturing firms is enhanced by reducing fixed costs. Land costs represent a large share of fixed costs. Leased industrial zones, compared to purchased ones, contribute significantly to reductions in fixed costs. This paper found that the segment of the manufacturing industry linked to labour wages is "leased" industrial zones. In other words, in a manufacturing agglomeration policy, "leased" industrial zones represent the accelerator segment of the sequence following the master switch.

Choosing to purchase an industrial zone requires greater fixed costs and capital investment. There are also higher sunk costs (costs already invested) after the purchase. This also increases risk. Risks such as future fluctuations in demand and contractions of the manufacturing sector are assumed. For these reasons, the use of leased industrial zones is important for risk avoidance and flexibility in manufacturing agglomerations.

The analysis in this paper has led to the identification of key considerations for the implementation of successful industrial agglomeration policies. First, it is essential to take into account the economies of sequence in the implementation of such policies. Second, the construction of an accelerator, which this paper finds as an example of economies of sequence, enhances the efficiency of policy implementation. Third, the construction of an industrial zone for rent is particularly recommended as a policy.

The following are the remaining tasks to be examined. First, the effectiveness of industrial parks in reducing fixed costs identified in this paper requires empirical evidence from other cases. Second, human capital is considered an important fixed-cost item that should be reduced. Instead of the new trade theory of spatial economics adopted in this paper, a model that considers human resource wages as a fixed cost in the new economic geography of spatial economics could be used. There remains room to apply other spatial economics models to sequence economics. Third, the "master switch" and "accelerator segments" were analysed theoretically and empirically as an economy of sequence in the segment construction process. In addition, there may be "stop" segments, i.e., segments that halt the segment construction process. One example is the institutional segment. If institutions are not in place or the operation of the institutions is not clear, firms may stop investing.

The World Bank (1993) and Wei (2020) concluded that industrial policies can only be applied to other countries if they satisfy certain conditions, although those conditions vary from country to country and are not easy to meet. Industrial agglomeration policies have been generally applied in many East Asian countries since the 1980s, and, although it cannot be denied that there have been many failures, they have achieved a certain degree of success and led to high economic growth in Asia, as has been widely reported.

However, Oqubay (2020) concludes that it is important to note that unevenness and divergence have been key features of the development of industrial zones, implying that there is no standard prescription. Oqubay and Lin (2020) encourage future research on industrial zones, and the editors emphasise the importance of interdisciplinary research and knowledge sharing. It should be noted that further research is needed on sequencing economics for industrial agglomeration policies from a more applicable regional development perspective.

**Funding:** This research received no external funding.

**Informed Consent Statement:** Not applicable.

**Data Availability Statement:** Publicly available datasets were analyzed in this study. This data can be found here: http://www.stats.gov.cn/tjsj/ndsj/2021/indexch.htm (Accessed on 9 June 2022).

**Acknowledgments:** We would like to thank Masahisa Fujita and Katsumi Nakayama for their comments on the draft of this paper.

**Conflicts of Interest:** The author declares no conflict of interest.

## Appendix A

**Table A1.** Data from Granger Causality Analysis.

| r | y | g | i | x | m | f | F22 | a |
|---|---|---|---|---|---|---|---|---|
| 1987 | 10.9 | 13.7 | 39.7 | 34.9 | 4.3 | 23 | 23 | 4.5 |
| 1988 | 11.3 | 14.5 | 39.6 | 18.2 | 27.4 | 32 | 32 | 39.1 |
| 1989 | 4.3 | 3.8 | 36.8 | 5.3 | 5.3 | 34 | 34 | 6.3 |
| 1990 | 3.9 | 3.2 | 35.2 | 19.2 | −13.3 | 35 | 35 | 2.9 |
| 1991 | 9.3 | 13.3 | 34.7 | 14.4 | 18.5 | 44 | 44 | 25.7 |
| 1992 | 14.2 | 21.7 | 36.2 | 18.1 | 28.3 | 110 | 110 | 150 |
| 1993 | 13.5 | 20.7 | 43.4 | 8.8 | 34.1 | 275 | 275 | 150 |
| 1994 | 11.8 | 17.4 | 40 | 35.6 | 10.4 | 337 | 337 | 22.5 |
| 1995 | 10.2 | 13.6 | 41.2 | 24.9 | 15.5 | 375 | 375 | 11.3 |
| 1996 | 9.7 | 12.3 | 39.6 | 1.5 | 5.1 | 416 | 416 | 10.9 |
| 1997 | 9 | 11.1 | 39.8 | 10 | 15.5 | 417 | 417 | 0.2 |
| 1998 | 7.8 | 8.9 | 37.7 | 3.3 | 0.5 | 454 | 454 | 8.9 |
| 1999 | 7.1 | 8.1 | 37.4 | 2.1 | 6.1 | 403 | 403 | −11.2 |
| 2000 | 8 | 9.4 | 36.3 | 1.9 | 27.9 | 407 | 407 | 1 |
| 2001 | 7.5 | 8.4 | 38.5 | 1.5 | 6.8 | 468 | 468 | 15 |
| 2002 | 9.1 | 9.8 | 37.9 | 22.4 | 21.3 | 527 | 527 | 12.6 |
| 2003 | 10 | 12.7 | 41.2 | 34.6 | 39.8 | 535 | 535 | 1.5 |
| 2004 | 10.1 | 11.1 | 43.3 | 35.4 | 35.8 | 606 | 606 | 13.3 |
| 2005 | 10.4 | 11.7 | 43.6 | 28.5 | 17.6 | 724 | 724 | 19.5 |
| 2006 | 11.6 | 13 | 43.6 | 27.2 | 19.7 | 727 | 727 | 0.4 |
| 2007 | 13 | 14.7 | 41.7 | 25.8 | 20.3 | 835 | 835 | 14.9 |
| 2008 | 9.6 | 9.8 | 42.5 | 17.6 | 18.7 | 1083 | 1083 | 29.7 |
| 2009 | 8.7 | 9.5 | 45.8 | −16.1 | −11.2 | 940 | 940 | −13.2 |
| 2010 | 10.4 | 12.3 | 0 | 31.4 | 39.1 | 1147 | 1147 | 22 |
| 2011 | 9.3 | 10.3 | 0 | 20.4 | 25.1 | 1239 | 1239 | 8 |
| 2012 | 7.9 | 8.4 | 0 | 9.2 | 5.2 | 1210 | 1210 | −2.3 |
| 2013 | 7.8 | 8 | 0 | 8.9 | 7.7 | 1239 | 1239 | 2.4 |
| 2014 | 7.3 | 7.4 | 0 | 4.4 | 1.1 | 1285 | 1285 | 3.7 |
| 2015 | 6.9 | 6.2 | 0 | −4.5 | −13.4 | 1355 | 1355 | 5.4 |
| 2016 | 6.7 | 6.3 | 0 | −7.2 | −4.2 | 1337 | 1337 | −1.3 |
| 2017 | 6.8 | 5.9 | 0 | 11.4 | 16 | 1363 | 1363 | 1.9 |
| 2018 | 6.6 | 5.8 | 0 | 9.1 | 16.2 | 1349 | 1349 | −1 |

y: GDP growth rata. g: Industry growth rate. i: domestic investment ratio (to GDP). x: export growth rate. m: import growth rate. Source: Asian Development Outlook, various years. Available online at https://www.adb.org/publications/series/asian%E2%88%92development%E2%88%92outlook (Accessed on 9 June 2022). f: inward foreign direct investment. a: growth rate of inward foreign direct investment. China Statistics Yearbook, National Bureau of Statistics, Available online at http://www.stats.gov.cn/tjsj/ndsj/2021/indexch.htm (Accessed on 9 June 2022).

**Table A2.** Date for Factor Analysis.

| | w1 | w2 | w3 | w4 | w5 | z1 | z2 | z3 | p1 | p2 | p3 | c1 | c2 |
|---|---|---|---|---|---|---|---|---|---|---|---|---|---|
| 1 | 636 | 984 | 2000 | 891 | 1782 | 103 | 3.09 | 20.5 | 0.075 | 0.76 | 0.48 | 1855 | 20,000 |
| 2 | 506 | 822 | 1268 | 1037 | 2185 | 96 | 2.79 | 22 | 0.095 | 0.68 | 0.46 | 804 | 13,915 |
| 3 | 669 | 1239 | 1865 | 1155 | 2530 | 218 | 6.96 | 29 | 0.095 | 0.75 | 0.61 | 1000 | 16,500 |
| 4 | 705 | 868 | 1435 | 1022 | 1928 | 46 | 3.53 | 20 | 0.15 | 0.83 | 0.53 | 750 | 3092 |
| 5 | 1124 | 1304 | 2509 | 1441 | 2973 | 203 | 6.49 | 43 | 0.1 | 0.59 | 0.5 | 900 | 24,000 |
| 6 | 595 | 1122 | 1601 | 1499 | 2968 | 660 | 2.18 | 28 | 0.15 | 0.58 | 0.68 | 900 | 16,000 |
| 7 | 572 | 903 | 1606 | 1240 | 2300 | 73.5 | 3.48 | 17.5 | 0.1 | 0.54 | 0.48 | 911 | 18,089 |
| 8 | 669 | 1125 | 1811 | 1001 | 1613 | 59 | 4.63 | 12 | 0.09 | 0.7 | 0.37 | 1546 | 20,000 |
| 9 | 1389 | 1856 | 3161 | 1576 | 3199 | 199 | 5.8 | 106 | 0.12 | 1.43 | 0.47 | 1250 | 16,000 |
| 10 | 2199 | 2138 | 4027 | 2506 | 4366 | 16035 | 25.5 | 80.5 | 0.15 | 0.995 | 1.8 | 880 | 19,000 |
| 11 | 1363 | 1725 | 2419 | 1658 | 2802 | 4152 | 9.83 | 14 | 0.1 | 0.355 | 0.37 | 900 | 16,000 |
| 12 | 225 | 483 | 1401 | 655 | 1437 | 47 | 2.73 | 5.02 | 0.05 | 0.57 | 1.12 | 1490 | 12,600 |
| 13 | 424 | 538 | 1320 | 572 | 1415 | 57 | 3.64 | 24 | 0.105 | 1.05 | 1.18 | 2500 | 12,200 |
| 14 | 277 | 546 | 1270 | 576 | 1440 | 51 | 3.38 | 10 | 0.09 | 1.46 | 1.21 | 1940 | 11,680 |
| 15 | 469 | 768 | 1677 | 722 | 1584 | 27.8 | 4.76 | 30.1 | 0.54 | 0.77 | 0.64 | 1420 | 12,450 |
| 16 | 281 | 516 | 1194 | 585 | 1644 | 49 | 5 | 26 | 0.08 | 0.18 | 1.18 | 1860 | 13,850 |
| 17 | 395 | 1335 | 3454 | 1390 | 3108 | 48 | 4.24 | 8.71 | 0.09 | 0.71 | 5.3 | 2300 | 1750 |
| 18 | 406 | 1804 | 7119 | 1660 | 6407 | 483 | 6.1 | 21 | 0.9 | 0.55 | 5.3 | 2770 | 2300 |
| 19 | 434 | 1969 | 3351 | 1588 | 3016 | 180 | 4.91 | 19 | 0.08 | 0.71 | 5.38 | 3070 | 850 |
| 20 | 480 | 1969 | 5083 | 1505 | 4575 | 113 | 4.88 | 16 | 0.09 | 0.55 | 5.3 | 2680 | 1650 |
| 21 | 357 | 1588 | 1742 | 928 | 1568 | 47 | 5.03 | 13 | 0.08 | 1.52 | 5.3 | 3320 | 1550 |
| 22 | 590 | 1866 | 4021 | 765 | 3619 | 105 | 6.44 | 12 | 0.05 | 7.26 | 11 | 5000 | 950 |
| 23 | 351 | 1412 | 3237 | 1340 | 2914 | 62 | 4.88 | 14 | 0.93 | 3.38 | 5.3 | 3130 | 1700 |
| 24 | 348 | 1297 | 1433 | 478 | 1720 | 200 | 2.6 | 11.55 | 0.021 | 0.38 | 1.5 | 2584 | 1800 |
| 25 | 923 | 2857 | 3780 | 1002 | 4716 | 108 | 4.25 | 25.89 | 0.06 | 0.34 | 0.007 | 2200 | 2600 |
| 26 | 542 | 3043 | 3650 | 591 | 3386 | 612 | 2.65 | 14.9 | 0.1045 | 4.02 | 0.6367 | 3900 | 2400 |
| 27 | 482 | 2807 | 3514 | 528 | 3240 | 122.01 | 3.67 | 1.67 | 0.11 | 10.211 | 0.5145 | 4200 | 3700 |
| 28 | 535 | 3028 | 3672 | 584 | 3386 | 200.32 | 3.06 | 22.52 | 0.1311 | 11.2213 | 0.8155 | 4400 | 2900 |
| 29 | 1270 | 3089 | 6491 | 1203 | 6491 | 124 | 9.49 | 23 | 0.09 | 0.8 | 0.68 | 1300 | 1100 |
| 30 | 567 | 3221 | 3921 | 619 | 3619 | 1009 | 4.14 | 35.66 | 0.095 | 9.37 | 0.63 | 3900 | 2400 |
| 31 | 385 | 663 | 1884 | 744 | 1642 | 181 | 6.07 | 24.5 | 0.145 | 2.37 | 0.57 | 1764 | 4554 |
| 32 | 407 | 614 | 1353 | 590 | 1470 | 208 | 5.1 | 20.13 | 0.07 | 4.42 | 0.38 | 2300 | 4500 |
| 33 | 430 | 818 | 1649 | 941 | 2076 | 139 | 4.56 | 14 | 0.65 | 0.46 | 0.23 | 1244 | 2144 |
| 34 | 294 | 495 | 1051 | 516 | 1863 | 157 | 4.74 | 31 | 0.2 | 1.605 | 1.405 | 1450 | 2100 |
| 35 | 1905 | 2681 | 4195 | 2692 | 4722 | 32 | 2.64 | 71 | 0.16 | 1.93 | 0.16 | 800 | 1500 |

Source: See Table 7.

**Table A3.** Loadings of Factor Analysis.

| JBIC 2007~2022 | MR1 | MR3 | MR2 | MR4 |
|---|---|---|---|---|
| a | 0.147 | 0.136 | 0.531 | −0.126 |
| b | −0.374 | 0.282 | 0.699 | 0.274 |
| c | −0.336 | 0.191 | | 0.482 |
| d | −0.134 | 0.435 | −0.398 | 0.252 |
| e | 0.42 | 0.455 | −0.605 | |
| f | −0.134 | 0.424 | 0.571 | −0.346 |
| g | | 0.644 | 0.218 | |
| h | −0.25 | 0.824 | | |
| i | 0.208 | | | 0.595 |
| j | 0.404 | −0.314 | −0.618 | |
| k | −0.702 | −0.417 | | 0.13 |
| l | 0.855 | −0.124 | | |
| m | 0.749 | −0.191 | | 0.207 |
| n | 1.008 | 0.28 | −0.1 | 0.143 |
| o | 1.055 | | | 0.148 |
| p | 0.203 | 0.949 | | |
| q | 0.292 | 0.858 | 0.134 | −0.103 |
| r | 0.968 | −0.115 | 0.483 | |

**Table A3.** *Cont.*

| | MR1 | MR3 | MR2 | MR4 |
|---|---|---|---|---|
| SS loadings | 5.736 | 3.867 | 2.336 | 1.002 |
| Proportion Var | 0.319 | 0.215 | 0.13 | 0.056 |
| Cumulative Var | 0.319 | 0.533 | 0.663 | 0.719 |
| **JETRO 2010 Asia** | **MR4** | **MR3** | **MR1** | **MR2** |
| w1 | 0.866 | −0.188 | −0.223 | 0.293 |
| w2 | 0.999 | | | |
| w3 | 0.807 | 0.296 | 0.186 | 0.116 |
| w4 | 0.56 | | | −0.3 |
| z1 | 0.735 | | 0.378 | −0.141 |
| z2 | 0.568 | 0.498 | 0.146 | |
| z3 | 0.683 | −0.244 | −0.326 | |
| p1 | −0.262 | −0.525 | 0.261 | −0.394 |
| p2 | −0.127 | | 0.173 | 0.974 |
| p3 | −0.288 | 1.029 | | −0.111 |
| c1 | | | 0.961 | 0.175 |
| c2 | | −0.287 | 0.655 | 0.654 |
| | **MR4** | **MR3** | **MR1** | **MR2** |
| SS loadings | 4.215 | 1.869 | 1.811 | 1.792 |
| Proportion Var | 0.351 | 0.156 | 0.151 | 0.149 |
| Cumulative Var | 0.351 | 0.507 | 0.658 | 0.807 |
| **JETRO 2021–22 World** | **MR1** | **MR3** | **MR2** | |
| w1 | 1.007 | 0.111 | −0.111 | |
| w2 | 0.447 | 0.717 | 0.175 | |
| w3 | 0.382 | 0.292 | 0.716 | |
| w4 | 0.725 | −0.289 | 0.398 | |
| w5 | 0.48 | 0.201 | 0.687 | |
| z2 | 0.833 | | | |
| z3 | 0.639 | | −0.183 | |
| p1 | −0.109 | −0.262 | 0.565 | |
| p2 | | 0.96 | −0.293 | |
| p3 | −0.18 | | 0.477 | |
| c1 | −0.323 | 0.812 | | |
| c2 | 0.508 | −0.268 | −0.303 | |
| | **MR1** | **MR3** | **MR2** | |
| SS loadings | 3.627 | 2.471 | 1.945 | |
| Proportion Var | 0.302 | 0.206 | 0.162 | |
| Cumulative Var | 0.302 | 0.508 | 0.67 | |

Note: fa (p33, nfactors = X, fm = "ml", rotate = "promax") by Program R. X = Number of eigne values greater than 1. Source: Author's calculation.

**Table A4.** High score cities of Factor Analysis.

| 2010 Asia | ML4 |
|---|---|
| 1. Beijing | 1.758 |
| 2. Shanghai | 1.1933 |
| 7. Shinzhen | 0.693 |
| 8. Bangkok | 0.8087 |
| 12. Bengaluru | 0.5813 |
| **2021–22 World** | **ML1** |
| 5. Shanghai | 1.3888 |
| 9. Beijing | 1.9823 |
| 10. Hong Kong | 3.9729 |
| 11. Taiwan | 1.8993 |
| 25. Buenos Aires | 0.8725 |
| 29. San Paulo | 1.8692 |

Source: Author's calculation.

**Appendix B**

*The Model*

First, the model obtains the first−order condition of the following problem:

$$\text{Minimise } \sum_{s=1}^{2} \int_{0}^{n_s} p_{sk}(i) m_{sk}(i) di,$$

$$\text{subject to } M_k = \left[ \sum_{s=1}^{2} \int_{0}^{n_s} m_{sk}(i)^{\frac{\sigma-1}{\sigma}} di \right]^{\frac{\sigma}{\sigma-1}}, \text{ for } k = 1, 2,$$

where $M_k$ is a substitution function defined over a continuum of varieties of goods consumed, $m_{sk}$ is the consumption of goods, and the parameter $\sigma$ describes the elasticity of substitution between any two of the varieties of goods. The number of differentiated products, or firms, in country k is given as $n_k$.

The utility function of a representative skilled worker in country k is given as

$$U_k = M_k{}^\mu A_k{}^{1-\mu}, \qquad \text{for } k = 1, 2, \tag{A1}$$

where $A_k$ is the agricultural consumption for a skilled worker living in country k. The respective income constraints for representative skilled workers in country 1 and country 2 are

$$y_k = \sum_{s=1}^{2} \int_{0}^{n_s} p_{sk}(i) m_{sk}(i) \, di + A_k, \ k = 1, 2, \tag{A2}$$

where $p_{sk}(i)$ is the price of the goods i produced in country s and consumed in country k.

Then, by maximising the utility (A1), the model obtains

$$m_{sk}(i) = (p_{sk}(i)^{-\sigma}/P_k{}^{1-\sigma}) y_k \, \mu, \text{ for } k = 1, 2, s = 1, 2, \tag{A3}$$

where the price index is

$$P_k = \left[ \sum_{s=1}^{2} \int_{0}^{n_s} p_{sk}(i)^{1-\sigma} \, di \right]^{1/(1-\sigma)}, \text{ for } k = 1, 2. $$

Next, consider a particular firm producing a specific variety of goods in country k (=1, 2). The firm trades one specific type of goods and incurs variable costs c and a fixed cost $F_k$. A firm producing variety i in country k maximises profits as follows:

$$\pi_k(i) = (p_{kk}(i) - c) \, m_{kk}(i) \, L_k + (p_{ks}(i) - \tau c) \, m_{ks}(i) \, L_s - F_k, \text{ for } k = 1, 2, s = 1, 2, s \neq k, \tag{A4}$$

where $\tau$ is the "iceberg" form of transport costs. The first−order condition gives the following equilibrium price:

$$p_{kk}(i) = p = \sigma c/(\sigma - 1), \ p_{ks}(i) = \tau p, \text{ for } k = 1, 2, s = 1, 2, s \neq k. \tag{A5}$$

The price index is

$$P_k = p(n_k + n_s \varphi)^{1/(1-\sigma)}, \text{ for } k = 1, 2, s = 1, 2, s \neq k, \tag{A6}$$

where $\varphi \equiv \tau^{(1-\sigma)}$.

Substituting (A3), (A5), and (A6) into (A4) obtains the firm's profits in regions 1 and 2:

$$\pi_k(i) = \pi_{kk}(i) + \pi_{ks}(i) - F_k = (\mu/\sigma) \left[ (y_k L_k)/(n_k + \varphi n_s) + (\varphi y_s L_s)/(\varphi n_k + n_s) \right] - F_k.$$

## Notes

[1]     See Oqubay and Lin (2020) at p. 10 and p. 28. Kuchiki (2020) defined the concept of "economies of sequence.".

[2]     According to Kou and Zhang (2020), by the end of 2018, there were 2447 industrial hubs in China, including 218 national ETDZs, 153 national HIDZs, and 110 other national development zones.

[3]     See Council of Local Authorities for International Relations (2003).

[4]     It should be noted that here, for example, the corporate tax rate was reduced from 30% to 15% in "state−level" development zones but not in "provincial−level' development zones.

[5]     The "industry−specific" and "region−specific" policies were implemented during this period (Hong Kong Grand Gazette, 19 June 1992).

[6]     The government emphasised the importance of Shanghai, with three major projects implemented in the 1990s. The first was the development of Pudong in Shanghai; the second was the construction of the Three Gorges Dam; and the third was the development of high−tech industries via HTDZs.

[7]     We provide an explanation of the conditions for what we refer to as the "turning point"' in the Chinese economy. Consider the industry sector and the agriculture sector in China, with a fixed wage regarded as a subsistence−level wage. Suppose that there is surplus labour in the agriculture sector. A firm can employ an unlimited number of people on the subsistence−level wage, and this situation is regarded as an unlimited labour supply. However, as the industry sector develops, it hires labour from the agriculture sector and the surplus labour in the agriculture sector disappears. Thus, it becomes impossible to hire at subsistence−level wages. The industrial sector can only hire at higher than subsistence−level wages and the minimum wage starts to rise. The point at which the industry sector can no longer hire at subsistence−level wages without restriction so that wages start to rise is called the "turning point".

[8]     This scientific development perspective is a guiding ideology launched in 2004 by the Hu Jintao Government of China, which took office in 2003. It aims for all−round, balanced, and sustainable development from a scientific and rational perspective, with people as the fundamental factor. It also involves a unified five−point plan on the basic premise of high economic growth, comprising urban and rural development, regional development, economic and social development, development in harmony with people and nature, and domestic development and opening−up to the outside world. The Chinese Government changed the central issues from a growth−oriented approach to one that sought to narrow the income gap and solve environmental problems while upgrading industrial infrastructure.

[9]     In the first phase, the Shanghai Pilot Free−Trade Zone aimed to act as a base at which to agglomerate various modern service industries (Tang Wenhong, Director General, Foreign Investment Bureau, Ministry of Commerce of China, Xinhua, 29 May 2019). Six industries were designated as "Modern Service Industries", with the financial industry the major one, followed by the aviation and transport service industry, commerce and trade services, professional services, culture and content, and social services. It is a bonded zone that is exempt from import duties on equipment and raw materials.

[10]     In the second phase, pilot free−trade zones were established in three regions in 2015—Guangdong, Tianjin, and Fujian—and in seven regions in 2017—Liaoning, Zhejiang, Henan, Hubei, and Sichuan, and Shaanxi provinces, and Chongqing. As part of the third phase, implemented in 2018, Hainan Province was designated as the 12th pilot free−trade zone; with pilot free−trade zones later established in six provinces and autonomous regions in 2019—Shandong, Jiangsu, Guangxi, Hebei, Yunnan, and Heilongjiang.

[11]     Note that industrial zones are not sold but rented in Cambodia, Vietnam, and Myanmar.

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
