# Peer review of "Accelerator for Agglomeration in Sequencing Economics: “Leased” Industrial Zones"

_economies, doi:10.3390/economies11120295_

Round 1
Reviewer 1 Report
Comments and Suggestions for Authors
The introduction provides a decent background on the significance of industrial zones in economic agglomeration policies, particularly referencing China's model. It integrates relevant references like Fujita, Krugman, and Venables (1999) and more recent works by Kuchiki, Oqubay and Lin (2020). However, while it lays a comprehensive groundwork for the paper's topic, it could benefit from a broader literature review that includes a variety of models beyond the Asian context to strengthen its relevance in the global context.
The cited references are pertinent to the research, especially in spatial economics and the role of industrial zones in economic growth. Nonetheless, incorporating additional perspectives from other regions would enhance the depth of the literature review and provide a more global perspective on the subject matter.
The paper's research design seems appropriate for investigating the role of "leased" industrial zones as accelerators in manufacturing agglomerations. It employs a chronological analysis of China’s policies and a factor analysis to identify drivers of investment, which are suitable methods for the research questions posed.
The methods are adequately described, including factor analysis to explore the relationship between fixed costs, particularly rents for "leased" industrial zones, and the agglomeration of manufacturing industries. The paper could improve by providing more detailed explanations of the factor analysis technique and justifying the choice of variables included in the analysis.
The results are clearly presented, with tables and figures supporting the findings. The factor analysis outcomes are discussed, highlighting the significance of "leased" industrial zones in reducing fixed costs and promoting agglomeration. Additional discussion on the implications of these findings for policy and economic development would be beneficial.
The results support the conclusions, with the paper asserting that leased industrial zones are crucial as accelerator segments in the sequence following the master switch in agglomeration policy. The conclusions could be strengthened by discussing potential limitations and suggesting avenues for future research.
The focus on "leased" industrial zones as an accelerator segment is novel and provides an interesting addition to the literature on industrial agglomeration policies.
The content is significant, particularly for policymakers and scholars interested in economic development and industrial policy mechanics.
The article is well-structured, with clear sections, making the content accessible. Some minor improvements in language clarity and coherence could enhance the overall presentation.
The paper is scientifically sound, employing recognized analytical methods and grounding its arguments in established economic theories.
The topic interests readers in economics, regional development, and public policy, especially those focusing on industrial agglomeration and economic development in developing countries.
The paper has merit and contributes to our understanding of the factors contributing to successful industrial agglomeration policies.
Author Response
Dear Reviewer 1,
I would like to thank you for your comments. They helped me significantly to improve my paper.
My responses are as follows:
Reviewer 1: Comments and responses:
The introduction provides a decent background on the significance of industrial zones in economic agglomeration policies, particularly referencing China's model. It integrates relevant references like Fujita, Krugman, and Venables (1999) and more recent works by Kuchiki, Oqubay and Lin (2020). However, while it lays a comprehensive groundwork for the paper's topic, it could benefit from a broader literature review that includes a variety of models beyond the Asian context to strengthen its relevance in the global context.
Response: I add the following sentences.
Prior to the 1980s, Hirschman (1958) recommended fostering domestic industry by protecting domestic firms, but his strategy of unbalanced growth was introduced under the liberalization of international trade and investment after the 1980s. In "The East Asian Miracle," the World Bank (1993) called export-led policies adopted in Asia through export processing zones the "export push strategy". Markusen (1996) classified industrial districts into five categories, including Marshall industrial districts and Italian-type industrial districts. Oqubay and Lin (2020), in the introduction to “The Oxford Handbook of Industrial Hubs and Economic Development,” introduce the sequence of economics defined by Kuchiki, i.e., the flowchart approach to industrial agglomeration, and use it as the foundation for empirical and case study evidence obtained in Asia, Latin America, and Africa. However, the flowchart approach lacked theoretical background and quantitative analysis.
The cited references are pertinent to the research, especially in spatial economics and the role of industrial zones in economic growth. Nonetheless, incorporating additional perspectives from other regions would enhance the depth of the literature review and provide a more global perspective on the subject matter.
Response: I add the following sentences.
As of 2018, there are approximately 6,000 existing and planned industrial parks worldwide, nearly 90% of which are in developing countries, with Asia accounting for nearly 70% and Africa for 5% (see UNCTAD (2019)). Oqubay (2020) noted that industrial parks in Ethiopia have played an insignificant role in the past, but could play a larger role in the overall industrial development strategy in the future.
Pietrobelli (2020) analyses cluster development policies in Latin American countries as follows: cluster development policies in the Latin American subcontinent began to be implemented in the 2000s, and most of them were financed by international donor agencies, including the Inter-American Development Bank; for example, cluster results for São Paulo and Minas Gerais, Brazil, show positive and significant effects on employment, export probability, and export levels.
The paper's research design seems appropriate for investigating the role of "leased" industrial zones as accelerators in manufacturing agglomerations. It employs a chronological analysis of China’s policies and a factor analysis to identify drivers of investment, which are suitable methods for the research questions posed.
The methods are adequately described, including factor analysis to explore the relationship between fixed costs, particularly rents for "leased" industrial zones, and the agglomeration of manufacturing industries. The paper could improve by providing more detailed explanations of the factor analysis technique and justifying the choice of variables included in the analysis.
Response: I add the following sentences.
Factor analysis is used to extract common factors latent behind observed variables. Here, the observed data are the explained variable xi, the common factor is the explanatory variable f, and the part that cannot be explained by the common factor is the error term ui, which is the unique factor. The coefficient bij of the explanatory variable that represents the common factor is the factor loading (xi = bij f + ui); here, the factor loadings are obtained by multiplying the square root of the eigenvalues of the factor loading matrix by the eigenvector. 
The results are clearly presented, with tables and figures supporting the findings. The factor analysis outcomes are discussed, highlighting the significance of "leased" industrial zones in reducing fixed costs and promoting agglomeration. Additional discussion on the implications of these findings for policy and economic development would be beneficial.
Response: I add the following sentences.
Choosing to purchase an industrial zone requires greater fixed costs and capital investment. There are also higher sunk costs (costs already invested) after the purchase. This also increases risk. Risks such as future fluctuations in demand and contractions of the manufacturing sector are assumed. For these reasons, the use of leased industrial zones is important for risk avoidance and flexibility in manufacturing agglomerations.
The analysis in this paper has led to the identification of key considerations for the implementation of successful industrial agglomeration policies. First, it is essential to take into account the economies of sequence in the implementation of such policies. Second, the construction of an accelerator, which this paper finds as an example of economies of sequence, enhances the efficiency of policy implementation. Third, the construction of an industrial zone for rent is particularly recommended as a policy.
The results support the conclusions, with the paper asserting that leased industrial zones are crucial as accelerator segments in the sequence following the master switch in agglomeration policy. The conclusions could be strengthened by discussing potential limitations and suggesting avenues for future research.
Response: I add the following sentences.
The following are the remaining tasks to be examined. First, the effectiveness of industrial parks in reducing fixed costs identified in this paper requires empirical evidence from other cases. Second, human capital is considered an important fixed-cost item that should be reduced. Instead of the new trade theory of spatial economics adopted in this paper, a model that considers human resource wages as a fixed cost in the new economic geography of spatial economics could be used. There remains room to apply other spatial economics models to sequence economics. Third, the "master switch" and "accelerator segments" were analysed theoretically and empirically as an economy of sequence in the segment construction process. In addition, there may be "stop" segments, i.e., segments that halt the segment construction process. One example is the institutional segment. If institutions are not in place or the operation of the institutions is not clear, firms may stop investing.
The focus on "leased" industrial zones as an accelerator segment is novel and provides an interesting addition to the literature on industrial agglomeration policies. The content is significant, particularly for policymakers and scholars interested in economic development and industrial policy mechanics. The article is well-structured, with clear sections, making the content accessible. Some minor improvements in language clarity and coherence could enhance the overall presentation. The paper is scientifically sound, employing recognized analytical methods and grounding its arguments in established economic theories. The topic interests readers in economics, regional development, and public policy, especially those focusing on industrial agglomeration and economic development in developing countries. The paper has merit and contributes to our understanding of the factors contributing to successful industrial agglomeration policies.
Response: I finished the proofreading of my paper.

Reviewer 2 Report
Comments and Suggestions for Authors
Line: "The purpose of this paper is to examine what the accelerator segment is after the master switch that reduces transportation costs when building manufacturing agglomer-ation segments. " it is not clear the sentence
This sentences is not clear in terms of idea: "We identify the priority of sequencing the segments of an agglomeration from the following three perspectives. First, a chronology of China's industrial agglomer-ation policies is used to identify the implementation of industrial agglomeration policies through the introduction of foreign capital. Second, this paper uses the conclusion drawn by the spatial economic model that the number of firms in an industrial agglomeration is inversely proportional to its fixed costs. Third, factor analysis reveals that both the wages of manufacturing workers and the rents of "leased" industrial zones belong to the same primary factor that leads to the agglomeration of manufacturing industries. "
What do you want to demonstrate? Is it possible to better the explain why those factor have been analysed? Why those factors instead of others? Is this part linked to Methodology part?
"The Japan Bank for International Cooperation (JBIC) conducted a study on the prom-ising reasons countries face in investment by Japanese companies. This paper presents the factors that promote investment through factor analysis. The main promising reasons are identified as factors that promote conditions for industrial agglomeration in each country. Factors related to incentives for foreign direct investment (FDI) include preferential tax policies for investment and stable policies to attract foreign investment. "
In this part source is missing.
"The Japan External Trade Organization (JETRO) surveyed Japanese companies oper-ating in 100 cities and regions in about 60 countries around the world to determine the investment-related costs of establishing operations in each city. The relationship between the wages of manufacturing workers and the rent of industrial zones is crucial to the de-cision to introduce FDI. Factor analysis indicates that reduction of fixed costs is a factor that accelerates the expansion of the number of firms attracting FDI. Therefore, when building manufacturing agglomerations, it is effective to establish " leased" industrial zones that utilize workers. "
In this part source is missing
"As shown in Figure 1, the accelerator is the "leased" industrial zone, while the master switch of manufacturing agglomeration policy is the reduction of transportation costs and the production of differentiated products. "
Not clear the accelerator, maybe needed to be identified at the beginning of the paper.
IMRAD structure is not respected, it could help in your case because the part of Introduction, Method, Discussion are not clear even if the experimental part is quite good. I advice to proof read the doc in English and to re-structure the paper.
Comments on the Quality of English Language
English needs a full review in every part, especially in the introduction, often used terms and concepts that do not respond to the scientific meaning of the idea.
Author Response
Dear Reviewer 2,
I would like to thank you for your comments. They helped me significantly to improve my paper.
My responses are as follows:
Reviewer 2: Comments and responses:
Line: "The purpose of this paper is to examine what the accelerator segment is after the master switch that reduces transportation costs when building manufacturing agglomeration segments. " it is not clear the sentence.
Response: I rewrite our purpose including the definition of accelerator as follows:
The purpose of this paper is to find the accelerator segments that comprise an industrial agglomeration. An accelerator segment is defined as a segment that increases the number of firms in an industrial agglomeration.
Thes sentences are not clear in terms of idea: "We identify the priority of sequencing the segments of an agglomeration from the following three perspectives. First, a chronology of China's industrial agglomeration policies is used to identify the implementation of industrial agglomeration policies through the introduction of foreign capital. Second, this paper uses the conclusion drawn by the spatial economic model that the number of firms in an industrial agglomeration is inversely proportional to its fixed costs. Third, factor analysis reveals that both the wages of manufacturing workers and the rents of "leased" industrial zones belong to the same primary factor that leads to the agglomeration of manufacturing industries. "
What do you want to demonstrate?
Is it possible to better the explain why those factors have been analysed? Why those factors instead of others?
Response: I changed the sentences as follows:
Our research methodology is provided as follows. First, China is a country that has successfully pursued a policy of industrial agglomeration through the introduction of foreign capital. Therefore, we focus on the chronology of industrial agglomeration in China. The results of the Granger causality test for China reveal the process of industrial agglomeration formed by the introduction of foreign capital. We find that inward foreign direct investment (FDI) is associated with industrial agglomeration policy. Between 1987 and 2009 in China, the increase in the domestic investment to GDP ratio, according to Granger causality, resulted in rises in the rates of import, industrial, and GDP growth. Similarly, inward FDI, based on Granger causality, resulted in rises in the rates of import, industrial, and GDP growth. The inward FDI growth rate can explain the industrial growth rate and GDP growth rate, with the inward FDI growth rate having a positive regression relationship with both rates.
After the Plaza Accord of 1985, the exchange rates of countries such as Japan and South Korea were revalued. As domestic costs rose, these countries shifted their production bases overseas, and foreign direct investment in Asia surged. This paper focuses on the outward investment behaviour of Japanese firms. Using a study conducted by the Japan Bank for International Cooperation (JBIC), we use factor analysis to identify the factors that promote investment. The most promising factors are those that promote conditions for industrial agglomeration in each country. We find that factors related to incentives for foreign direct investment (FDI) include preferential tax policies for investment and stable policies to attract foreign investment.
Second, we apply a two-region model of the new economic geography in spatial economics to obtain a conclusion that the number of firms moving to a region is inversely proportional to the fixed costs of that region. This conclusion is the rationale for the fact that the accelerator segment is a fixed-cost factor of production.
Third, among the fixed costs, those related to the industrial agglomeration of labour-intensive manufacturing industries were identified through factor analysis using a survey conducted on Japanese companies by JETRO in 100 cities and regions in around 60 countries to determine investment-related costs when establishing operations in each city. We show that the rent of leased industrial zones was included in the same factor as the wages of workers in labour-intensive manufacturing industries. Therefore, we conclude that the establishment of leased industrial zones reduces fixed costs and increases the number of firms in industrial agglomerations.
Is this part linked to Methodology part?
Response: Yes. I add the following sentence as above.
Our research methodology is provided as follows.
"The Japan Bank for International Cooperation (JBIC) conducted a study on the promising reasons countries face in investment by Japanese companies. This paper presents the factors that promote investment through factor analysis. The main promising reasons are identified as factors that promote conditions for industrial agglomeration in each country. Factors related to incentives for foreign direct investment (FDI) include preferential tax policies for investment and stable policies to attract foreign investment. "
In this part source is missing.
Response: I changed the sentences as follows:
Using a study conducted by the Japan Bank for International Cooperation (JBIC), we use factor analysis to identify the factors that promote investment. The most promising factors are those that promote conditions for industrial agglomeration in each country. We find that factors related to incentives for foreign direct investment (FDI) include preferential tax policies for investment and stable policies to attract foreign investment.
"The Japan External Trade Organization (JETRO) surveyed Japanese companies operating in 100 cities and regions in about 60 countries around the world to determine the investment-related costs of establishing operations in each city. The relationship between the wages of manufacturing workers and the rent of industrial zones is crucial to the decision to introduce FDI. Factor analysis indicates that reduction of fixed costs is a factor that accelerates the expansion of the number of firms attracting FDI. Therefore, when building manufacturing agglomerations, it is effective to establish " leased" industrial zones that utilize workers. "
In this part source is missing
Response: I changed the sentences as follows:
Third, among the fixed costs, those related to the industrial agglomeration of labour-intensive manufacturing industries were identified through factor analysis using a survey conducted on Japanese companies by JETRO in 100 cities and regions in around 60 countries to determine investment-related costs when establishing operations in each city. We show that the rent of leased industrial zones was included in the same factor as the wages of workers in labour-intensive manufacturing industries.
"As shown in Figure 1, the accelerator is the "leased" industrial zone, while the master switch of manufacturing agglomeration policy is the reduction of transportation costs and the production of differentiated products. "
Not clear the accelerator, maybe needed to be identified at the beginning of the paper.
Response: I add the following sentences.
The purpose of this paper is to find the accelerator segments that comprise an industrial agglomeration. An accelerator segment is defined as a segment that increases the number of firms in an industrial agglomeration.
IMRAD structure is not respected, it could help in your case because the part of Introduction, Method, Discussion are not clear even if the experimental part is quite good. I advice to proof read the doc in English and to re-structure the paper.
Response: I ask the editing service of MDPI. I changed the introduction as follows:
Industrial agglomeration policy refers to the creation of agglomerations through policy. According to Fujita, Krugman, and Venables (1999), agglomeration means the clustering of economic activities that are created and maintained through some form of circular logic.
Industrial agglomeration policies have been adopted in East Asia since the 1980s. The development of industrial zones brings together economic agglomeration and industrial clusters of economic activity. The prototype of industrial zones in Asia is the export processing zone concept in Kaohsiung, Taiwan, established in 1965. This model was then introduced in Penang, Malaysia, in the 1970s, and in Tan Thuan, Ho Chi Minh City, in the 1990s.
Fujita, Krugman, and Venables (1999) established spatial economics, or the study of where economic activity takes place and why. According to Oqubay and Lin (2020), the number of industrial zones in Asia has increased dramatically since the 1980s, with the Asia–Pacific region alone accounting for over 65% of global employment and exports. UNCTAD (2019) notes that in China, special economic zones (SEZs) have a strong positive effect on foreign direct investment (FDI), with SEZs accounting for more than 80% of cumulative FDI. Oqubay and Lin (2020) showed, through a dynamic approach, that understanding industrial hubs is important for the production-centred paradigm.
As of 2018, there are approximately 6,000 existing and planned industrial parks worldwide, nearly 90% of which are in developing countries, with Asia accounting for nearly 70% and Africa for 5% (see UNCTAD (2019)). Oqubay (2020) noted that industrial parks in Ethiopia have played an insignificant role in the past, but could play a larger role in the overall industrial development strategy in the future.
Pietrobelli (2020) analyses cluster development policies in Latin American countries as follows: cluster development policies in the Latin American subcontinent began to be implemented in the 2000s, and most of them were financed by international donor agencies, including the Inter-American Development Bank; for example, cluster results for São Paulo and Minas Gerais, Brazil, show positive and significant effects on employment, export probability, and export levels.
Prior to the 1980s, Hirschman (1958) recommended fostering domestic industry by protecting domestic firms, but his strategy of unbalanced growth was introduced under the liberalization of international trade and investment after the 1980s. In "The East Asian Miracle," the World Bank (1993) called export-led policies adopted in Asia through export processing zones the "export push strategy". Markusen (1996) classified industrial districts into five categories, including Marshall industrial districts and Italian-type industrial districts. Oqubay and Lin (2020), in the introduction to “The Oxford Handbook of Industrial Hubs and Economic Development,” introduce the sequence of economics defined by Kuchiki, i.e., the flowchart approach to industrial agglomeration, and use it as the foundation for empirical and case study evidence obtained in Asia, Latin America, and Africa. However, the flowchart approach lacked theoretical background and quantitative analysis.
Agglomeration segments are classified into four categories: physical infrastructure, institutions, human resources, and living environment, as shown in Table 1. Kanai and Ishida (2000) emphasised the importance of the cumulative process because, in spatial economics, the construction of any segment of agglomeration takes “time” in addition to space. Kuchiki (2023) reflected on the analysis of the accumulation process as follows. First, Fujita and Kuchiki (2006) applied the flowchart method to the construction of the cumulative process as shown in Figure 1. Second, Kuchiki (2020) proposed an architectural theory in the economy of sequence with respect to accumulation to find the optimal sequence for efficient segment construction. "Economies of sequence" in sequencing economics is defined as the ordering of any two segments in the set of segments that make up an agglomeration to efficiently construct that agglomeration. Third, Kuchiki (2023) used the fact that spatial economics models derive segments that satisfy the symmetry breaking condition to find that segments related to transport costs are the "master switch" for ordering segments of urban agglomerations. When a stable symmetric equilibrium is broken, the construction of the segments of an agglomeration equilibrium begins. However, no study has examined what the accelerator segment next to the master switch is when constructing the segments of a manufacturing agglomeration.
The purpose of this paper is to find the accelerator segments that comprise an industrial agglomeration. An accelerator segment is defined as a segment that increases the number of firms in an industrial agglomeration.
Our research methodology is provided as follows. First, China is a country that has successfully pursued a policy of industrial agglomeration through the introduction of foreign capital. Therefore, we focus on the chronology of industrial agglomeration in China. The results of the Granger causality test for China reveal the process of industrial agglomeration formed by the introduction of foreign capital. We find that inward foreign direct investment (FDI) is associated with industrial agglomeration policy. Between 1987 and 2009 in China, the increase in the domestic investment to GDP ratio, according to Granger causality, resulted in rises in the rates of import, industrial, and GDP growth. Similarly, inward FDI, based on Granger causality, resulted in rises in the rates of import, industrial, and GDP growth. The inward FDI growth rate can explain the industrial growth rate and GDP growth rate, with the inward FDI growth rate having a positive regression relationship with both rates.
After the Plaza Accord of 1985, the exchange rates of countries such as Japan and South Korea were revalued. As domestic costs rose, these countries shifted their production bases overseas, and foreign direct investment in Asia surged. This paper focuses on the outward investment behaviour of Japanese firms. Using a study conducted by the Japan Bank for International Cooperation (JBIC), we use factor analysis to identify the factors that promote investment. The most promising factors are those that promote conditions for industrial agglomeration in each country. We find that factors related to incentives for foreign direct investment (FDI) include preferential tax policies for investment and stable policies to attract foreign investment.
Second, we apply a two-region model of the new economic geography in spatial economics to obtain a conclusion that the number of firms moving to a region is inversely proportional to the fixed costs of that region. This conclusion is the rationale for the fact that the accelerator segment is a fixed-cost factor of production.
Third, among the fixed costs, those related to the industrial agglomeration of labour-intensive manufacturing industries were identified through factor analysis using a survey conducted on Japanese companies by JETRO in 100 cities and regions in around 60 countries to determine investment-related costs when establishing operations in each city. We show that the rent of leased industrial zones was included in the same factor as the wages of workers in labour-intensive manufacturing industries. Therefore, we conclude that the establishment of leased industrial zones reduces fixed costs and increases the number of firms in industrial agglomerations.
As shown in Figure 1, the accelerator is the "leased" industrial zone, while the master switches of a manufacturing agglomeration policy are the reduction in transportation costs and the production of differentiated products.
  The analysis in this paper suggests measures for successful industrial agglomeration policies. A consideration of the economies of sequence in sequencing economics is essential to the implementation of such policies. The construction of accelerators will increase the efficiency of policy implementation. Further research in sequencing economics will be completed in the future.
Comments on the Quality of English Language
English needs a full review in every part, especially in the introduction, often used terms and concepts that do not respond to the scientific meaning of the idea.
Response: I ask the editing service of MDPI.

Round 2
Reviewer 2 Report
Comments and Suggestions for Authors
All the comments have been tackled by the Authors. Now it is fine. Thank you
Comments on the Quality of English Language
All the comments have been tackled by the Authors. Now it is fine. Thank you. The review is effective.